# Comparison of the UEscope Video Laryngoscope with the Traditional Direct Laryngoscope in Neonates and Infants: A Randomized Clinical Trial

**DOI:** 10.3390/children9081161

**Published:** 2022-08-02

**Authors:** Min-Suk Chae, Jae-Hee Chung, Jung-Woo Shim, Jae-Sik Park, Jin-Hoon Bae, Hyung-Mook Lee

**Affiliations:** 1Department of Anesthesiology and Pain Medicine, Seoul St. Mary’s Hospital, College of Medicine, The Catholic University of Korea, Seoul 06591, Korea; shscms@catholic.ac.kr (M.-S.C.); tintinshim@gmail.com (J.-W.S.); jaesik.park@gmail.com (J.-S.P.); ccac11@naver.com (J.-H.B.); 2Department of Surgery, Seoul St. Mary’s Hospital, College of Medicine, The Catholic University of Korea, Seoul 06591, Korea; jhjung@catholic.ac.kr

**Keywords:** child, infant, intubation, laryngoscopes, newborn

## Abstract

The role of video laryngoscopy in adults is well established, but its role in children is still inconclusive. Previous studies on the UEscope in pediatric patients with difficult airways showed that it could reduce the time to intubation (TTI) compared to a conventional direct laryngoscope. The main objective of the current study was to investigate if the use of the UEscope could reduce the TTI in neonates and infants. Forty patients under 12 months old were recruited from a single tertiary hospital from March 2020 to September 2021 and were randomly assigned to the direct laryngoscope group (*n* = 19, neonates = 4, infants = 15) or UEscope group (*n* = 21, neonates = 6, infants = 15). Although the quality of glottic view was comparable in both groups, the TTI was significantly lower in the UEscope group in both the “intention-to-treat” (–19.34 s, 95% confidence interval = –28.82 to –1.75, *p* = 0.0144) and “as treated” (–11.24 s, 95% confidence interval: –21.73 to 0, *p* = 0.0488) analyses. The UEscope may be a better choice for tracheal intubation than conventional direct laryngoscope in neonates and infants.

## 1. Introduction

Since the first use of an endotracheal tube for drowned persons, endotracheal intubation has been considered a safe and definite way of securing the airway in both adults and children [1]. However, the airway management environment is complicated in neonates and infants. Although anesthesiologists pre-oxygenate neonates’ and infants’ lungs before intubation, only a few minutes are available before desaturation begins due to the smaller total lung capacity and higher oxygen consumption rate in these patients compared to those in older patients [2]. In addition, infants and children have more vulnerable anatomies, such as a more anterior and cephalad larynx, a large epiglottis and tongue, a shorter mandible, and a prominent occiput [3]. These physiologic and anatomic characteristics lead to an approximately seven times higher incidence of difficult laryngoscopy (Cormack and Lehane Laryngeal View [CL] Grade ≥ III) in neonates and infants compared to that in older children [4]. Even the normal airways of young children can be difficult to manage for anesthesiologists with limited pediatric experience [5].

In adults, new devices containing optical fibers or lenses with video screens have been introduced for tracheal intubation. These devices include, but are not limited to, hybrid devices, optical stylets, single-use fiberscopes, and fiberoptic tracheal tubes. Although well-designed randomized clinical trials (RCTs) are limited, the efficacy of these new technologies have been reported in various studies including case series and cohort studies [6]. However, in the pediatric population, direct laryngoscopy is still the mainstream tool of tracheal intubation for surgical procedures, while video laryngoscopy (VL) is increasingly popular and used successfully in difficult airway situations [7].

In 1990, VL was introduced to help in the application of endotracheal intubation, although the overall benefits of VL are still controversial in pediatric patients, especially neonates. In a meta-analysis of 14 RCTs, it was suggested that VL in children was associated with an improved glottic view when used in both normal airways and potentially difficult intubations. However, the time to intubation (TTI) was prolonged, and the incidence of failed intubation was increased significantly [8]. Another review of VL use in children excluding neonates suggested a similar conclusion, although it should be noted that the quality of evidence was low [9]. A review of VL use in neonates suggested that its use was associated with higher first-attempt success, but the TTI and the number of intubation attempts were not decreased. However, the intubations included in these studies were performed by trainees who did not have adequate experience [10]. A recent meta-analysis of 27 RCTs in children also showed that VL required a longer TTI, but the time was similar in infants. In the review, the use of VL improved the percentage of glottis opening (POGO) score and intubation trauma but did not lower the CL grade [11].

The UEscope (VL400, Zhejiang UE Medical Corp, Xianju, China) is a relatively new portable video laryngoscope with an adjustable video screen. It is supplied with a Miller #0 straight blade and a Macintosh #1 curved blade with a 40° intermediate angle curvature for neonates and infants [12]. In neonates and infants, studies comparing the UEscope with conventional direct laryngoscopy are rare. Previous meta-analyses did not include studies on the UEscope. Thus, in this study, we aimed to compare the UEscope with conventional direct laryngoscopy in aspects of the TTI; time to best view (TTBV); POGO score; CL grade; effects of the backward, upward, rightward, and posterior pressure (BURP) maneuver; first-time success rates; and incidence of desaturation during intubation in neonates and infants.

## 2. Materials and Methods

### 2.1. Study Population

In this prospective, randomized, parallel, single-blinded clinical trial, 40 neonates and infants who underwent elective general surgery in a single tertiary hospital between March 2020 and September 2021 were prospectively enrolled. Patients up to 12 months of age were included. Patients with a history of head and neck surgery or radiation therapy, possible congenital abnormality of the cervical vertebrae, and/or a history of tracheostomy, cricothyroidotomy, or intubation were excluded.

### 2.2. Study Protocol

During the study period, children were randomly assigned to two groups, namely the UEscope group (UE) and the conventional direct laryngoscope group (DL). Randomization was prepared by the contract research organization Medical Excellence (Seoul, South Korea). Medical Excellence provided 200 sealed envelopes with a study number and computer-generated randomized allocation information. The random block size was not shared with the authors. One of the co-authors stored them in a secure place and unsealed the appropriate envelope after the study participant was admitted to the operation room. The allocation information was “UEscope” for UE and “Conventional” for DL. The participants were allocated in a 1: 1 ratio. The participants were blinded to group allocation, but the anesthesiologist performing the intubation could not be blinded since the type of laryngoscope could not be hidden.

A donut-shaped headrest was used to support the head of all infants. The headrest was not used in neonates. All patients enrolled in the current study were anesthetized according to a standard protocol. All patients were sedated with 2 mg/kg of intravenous ketamine in the waiting room. After their transfer into the operating room, 100% oxygen was administered to the patients through a mask. The patients were then administered 2% inhalational sevoflurane along with an intravenous bolus of 0.67 µg/kg of remifentanil and 0.6 mg/kg of rocuronium. Before performing tracheal intubation, participants were manually ventilated with 100% oxygen for 2 min to ensure the full effects of the neuromuscular block.

A Miller #0 straight blade was used for neonates who were 30 days old or less. A Macintosh #1 curved blade was used for infants aged 30 days to 12 months. A stylet was used in all of the intubation attempts. The stylet was shaped to fit the curve of the appropriate blade. For the UEscope with a Macintosh #1 curved blade, a preformed stylet supplied with the UEscope was used.

Laryngoscopy with the UEscope was performed without displacement of the tongue [10]. Laryngoscopy with a conventional direct laryngoscope was conducted with the blade inserted in the right side of the mouth and displacing the tongue to the left side. In both groups, the tip of the blade was placed in the vallecula, and the epiglottis was elevated. With the signal of the anesthesiologist, an anesthesiology nurse performed the BURP maneuver. After securing the best view, the endotracheal tube was inserted with the stylet. The stylet was removed after the anesthesiologist confirmed the insertion of the tube.

The TTI was defined as the time interval from the laryngoscope passing the lip to confirming tracheal intubation, subtracting the time interval of the BURP maneuver. The TTBV was defined as the time from the laryngoscope passing the lip to confirming the best glottic view, while the TTBV with the BURP maneuver was defined as the time from the laryngoscope passing the lip to confirming the best glottic view after the BURP maneuver.

During intubation, all processes were recorded to ensure the correct time record. The glottic view was recorded using the built-in camera of the UEscope or fiberscope. When using a direct laryngoscope, an assistant inserted the flexible tracheal intubation fiberscope LF-DP(Olympus Medical, Center Valley, PA, USA) along the inferolateral border of the blade as directed by the anesthesiologist performing the intubation. The fiberscope was connected to the monitor in the operating room. The anesthesiologist performing the intubation confirmed that the view of the fiberscope was consistent with the visual observation. The fiberscopic view could differ from direct visualization, but the POGO score and CL grade did not change. Another pediatric anesthesiologist independently determined the POGO score and CL grade based on pictures, and two anesthesiologists discussed and confirmed the result.

If the intubation failed, the time interval for the next try was added to the overall time. The maximum limit was three attempts or 5 min. If intubation was not possible after the limit was reached, the laryngoscope was swapped to that of the other group, and intubation was tried again.

Intraoral bleeding was detected by visual observation and suctioning after the endotracheal tube was secured and after the patient was extubated.

To reduce interpersonal bias, a single pediatric attending anesthesiologist with more than 10 years of experience in pediatric anesthesia conducted all intubations. The anesthesiologist had performed more than 100 intubations in newborns and infants using the UEscope before the trial began.

### 2.3. Objectives

The primary objective of the current study was to determine if there was a reduction in the TTI using the UEscope compared to that when using the conventional direct laryngoscope for intubation in neonates and infants.

The secondary objectives were the TTBV with and without the BURP maneuver, the POGO score with and without the BURP maneuver, CL grade with and without the BURP maneuver, effects of the BURP maneuver on glottic view, number of attempts, rate of first attempt success, incidence of desaturation, lowest SpO_2_ during intubation, and presence of intraoral bleeding

### 2.4. Sample Size Calculation

A prior study assessing 240 children aged 2–10 years with normal airways showed that the TTIs (mean ± SD) were 31.7 ± 6.2 s in the DL group and 23.8 ± 4.5 s in the UE group [13]. Another study compared 80 children with normal airways and 30 children with difficult airways. In the normal airway group, the TTIs (mean ± SD) were 19.1 ± 6.0 s when using direct laryngoscopy (*n* = 40) and 16.5 ± 6.9 s when using the UEscope (*n* = 40), while in the difficult airway group, the TTIs (mean ± SD) were 24.2 ± 5.9 s when using direct laryngoscopy (*n* = 15) and 18.5 ± 4.8 s when using the UEscope (*n* = 15) [14]. Based on previous studies, a sample size of 20 per group was required to detect a 30% difference in TTI between the two groups with a two-sided test using α = 0.05 and β = 0.2, allowing for a 10% drop-out rate. However, to analyze the difference in TTI between subgroups, sample sizes of 20 per group in the neonate group and 30 per group in the infant group were required.

### 2.5. Statistical Analyses

GraphPad Prism version 7.05 for Windows (GraphPad Software, La Jolla, CA, USA, www.graphpad.com accessed on 2 August 2022) was used for the statistical analyses. All tests were two-sided and a *p*-value of less than 0.05 was statistically significant and marked with an asterisk (*).

Results were presented as the mean ± standard deviation or the median (interquartile range), as appropriate. Calculation of the difference between median and the 95% confidence interval (CI) of the difference between median was based on the Hodges–Lehmann method. Differences between the groups were analyzed using the student’s *t*-test or Mann–Whitney U test, when appropriate. The Wilcoxon matched-pairs signed-rank test was used to analyze differences in the POGO score and CL grade before and after performing the BURP maneuver in the same group. The D’Agostino–Pearson normality test was used to test for normal distribution.

### 2.6. Ethics

The current study was conducted according to the guidelines of the Declaration of Helsinki and complies with the Consolidated Standards of Reporting Trials (CONSORT) guidelines (Appendix A). The study protocol was reviewed and approved by the Institutional Review Board of St. Mary’s Hospital, Seoul, Korea (KC19DESI0621) on 4 November 2019 and was registered at the Clinical Research Information Service (http://cris.nih.go.kr; KCT0004676 accessed on 30 January 2020). The first participant was recruited on 30 March 2020. Informed consent was obtained from all patients’ parents or legal guardians.

## 3. Results

Forty-two patients were included in the study, but one patient dropped out due to a legal guardian-related issue, and one patient dropped out since the participant’s mother changed her mind after the procedure (Figure 1).

The baseline patient characteristics are presented in Table 1. More neonates were assigned to the UE than the DL (28.6% vs. 21.1%). The age of the youngest child was two days in both groups.

### 3.1. Primary Objective

The primary objective of the current study was to compare the TTI between the two groups. In DL, 2 patients were intubated using the UEscope after 5 min of trying the conventional laryngoscope. For the “intention-to-treat” analysis, the TTI was considered to be 300 s for these two participants. For the “as-treated” analysis, the two participants were moved from the DL group into the UE group. Data are presented as the median with interquartile range in Appendix B (Table A1).

#### 3.1.1. Intention-to-Treat Analysis of TTI

The TTI was shorter in the UE than in the DL, and the difference was statistically significant. The difference between median was –13.86 s, with a 95% CI of –28.82 to –1.75 (* *p*-value = 0.0144; Figure 2).

#### 3.1.2. As-Treated Analysis of TTI

The TTI was shorter in the UE than in the DL; and the difference was statistically significant. The difference between median was −10.37 s, with a 95% CI of −21.73 to 0 (* *p*-value = 0.0488; Figure 2)

### 3.2. Secondary Objectives

All analyses of the secondary objectives were intention-to-treat analyses (Appendix B. Table A2). The mean with a 95% CI was used for the secondary objectives to show differences between the two groups more clearly.

#### 3.2.1. TTBV

The TTBV without the BURP maneuver was shorter in the UE than in the DL, and the difference was statistically significant. The difference between means was −7.13 s, with a 95% CI of −14.1 to −0.16 (* *p*-value = 0.0217; Figure 3).

The TTBV with BURP maneuver was shorter in the UE than in the DL, and the difference was statistically significant. The difference between means was −7.30 s, with a 95% CI of −14.95 to −0.39 (* *p*-value = 0.0472; Figure 3).

#### 3.2.2. Best POGO Score

The best POGO score without the BURP maneuver was not statistically significant between the two groups. The difference between means was 0.50, with a 95% CI of −1.30 to 2.30 (*p*-value = 0.5780; Figure 4).

The best POGO score with the BURP maneuver was not statistically significant between the two groups. The difference between means was −0.40, with a 95% CI of −2.82 to 2.03 (*p*-value = 0.8977; Figure 4).

#### 3.2.3. CL Grade

The CL grade without the BURP maneuver was not statistically significant between the two groups. The difference between means was −0.12, with a 95% CI of −0.56 to 0.32 (*p*-value = 0.5868; Figure 5).

The CL grade with the BURP maneuver was not statistically significant between the two groups. The difference between means was −0.13, with a 95% CI of −0.60 to 0.35 (*p*-value = 0.7085; Figure 5).

#### 3.2.4. BURP Maneuver

The BURP maneuver was associated with a higher POGO score in both groups. The effect was statistically significant. In the DL, the difference between means was 2.9 (95% CI: 1.52 to 4.27, * *p*-value = 0.0002; Figure 6).

In the UE, the difference between means was 2 (95% CI: 1.2 to 2.8, * *p*-value < 0.0001; Figure 6).

The BURP maneuver was associated with a lower CL grade in both groups. The effect was statistically significant. In DL, the difference between means was−0.42 (95% CI: −0.75 to −0.09, * *p*-value = 0.0352; Figure 7).

The difference between means with or without the BURP maneuver in UE was −0.43 (95% CI: −0.66 to −0.20, **p*-value = 0.0009; Figure 7)

#### 3.2.5. Number of Intubation Attempts

The number of intubation attempts was lower in the UE than in the DL, and the difference was statistically significant. The difference between means was −0.54, with a 95% CI of −0.90 to −0.17 (* *p* = 0.0082; Figure 8).

#### 3.2.6. First-Attempt Success Rate

The first-attempt success rate was higher in the UE group than in the DL group (90.5% vs. 52.6%). The difference between means was −0.54 (95% CI: −0.9 to −0.1727, *p* = 0.0049).

#### 3.2.7. Incidence of Desaturation

Although the most severe desaturation happened in the UE group, the incidence of desaturation was higher in the DL group than in the UE group (36.8% vs. 9.5%). The difference was statistically significant. The difference between means was −0.27, with a 95% CI of −0.53 to −0.01 (* *p*-value = 0.0432). The lowest SpO2 level was 83% in the DL and 76% in the UE.

## 4. Discussion

The current study showed that the UEscope was associated with a shorter TTI, shorter TTBV, lower number of total intubation attempts, higher first-attempt success rate, and lower incidence of desaturation than those of a conventional direct laryngoscope in neonates and infants. However, the quality of the glottic view was comparable with that of the conventional direct laryngoscope. One possible explanation for this is the high incidence of difficult airways and the effects of the BURP maneuver [15,16]. The BURP maneuver was more effective with difficult airways in UE. Among the participants, 31.6% in the DL group and 38.1% in the UE group had a CL grade of III. After the BURP maneuver, 26.3% of participants in the DL group had a CL grade of III, but this decreased to 9.5% in the UE group, which is a 75% reduction. However, the change in POGO score before and after the BURP maneuver was only 2 out of 10 in both groups. In other words, although the view became more favorable after the BURP maneuver, the change was still small. This small change in the view made decisive effects on intubation with the UEscope. It seems that in difficult airways such as those of neonates and infants, it is much easier to conduct intubation with the UEscope with the help of the BURP maneuver, even if the initial glottic view is unfavorable.

The decrease in TTI was in line with the findings of previous studies, which showed that, in children with difficult airways, the UEscope provided a better glottic view, higher first-attempt success rate, and shorter TTI [14,17]. On the contrary, a recent metanalysis including 46 RCTs of videoscope showed that video laryngoscopes reduced the risk of failed first intubation attempts but did not decrease TTI in children, although every video laryngoscope had a different performance metrics [18].

This result could be partially explained by the characteristics of UEscope, which is lightweight (<200 g), relatively small (height < 20 cm), and equipped with an all-angle adjustable monitor on top of the handle that helps with hand-eye coordination [12] These features can help to reduce intubation time by providing more space for the handling of the endotracheal tube, especially in neonates and infants. The UEscope’s standard type blade is also helpful. Other videolaryngoscopes such as the Glidescope, McGrath, and Storz have a steep upward angled blade, while the UEscope #1 curved blade has a milder 40° angle. The curvature of the UEscope blade is more similar to that of the Macintosh direct laryngoblade than other videoscopes [12] In addition, a recent international, multicenter, RCT supports the use of videoscope with a standard type blade for neonate and infants [19]. In a few studies with patients in a critical situation, the UEscope outperforms other laryngoscopes. In a study comparing the UEscope and conventional direct laryngoscopy with a Miller’s #2 blade in a manikin simulating cardiopulmonary resuscitation with uninterrupted chest compressions in a 5-year-old child, the UEscope was associated with a shorter TTI, a higher first-attempt success rate, and better POGO score and CL grade [20]. In an RCT including 120 adult patients with cervical immobilization, TTBV was decreased and first-time success rate was higher with the UEscope compared to Glidescope VL [21].

Although a supraglottic airway device (SGA) was not used in the current study, a recently published guideline on the management of difficult airways encourages the use of an SGA in difficult airway situations [22]. However, in a recent large prospective observational study, SGAs were rarely used in failed tracheal intubations, although the success rate of SGA placement within two insertion attempts is more than 99% during pediatric anesthesia. SGA usage is increasing in non-emergent, outpatient, and non-surgical settings [7].

The current study has some limitations. First, only 40 participants were recruited, far fewer than the original 100 participants planned. At first, we planned to recruit 100 participants to conduct a subgroup analysis. However, during the study period, it became practically impossible to recruit 100 participants in our institution due to policy changes in the medical system. Thus, the study ended after 40 participants were recruited. Post-hoc analysis of the primary objective showed that the “as-treated” TTI comparison had a power of 84.6%. The number of participants was enough to analyze the primary objective, TTI, as anticipated. However, the small sample size could be a source of bias. Second, all of the participants were patients from a single tertiary hospital. There may be a selection bias, which presented as a high incidence of difficult airways. Third, patients with a previous history of intubation were excluded due to the risk of tracheomalacia. This led to the exclusion of patients with a very high risk of difficult airways, including premature neonates, which could affect the results. Fourth, we did not measure the neuromuscular blocking effects of rocuronium. Instead, we waited for 2 min before performing endotracheal intubation. In young children, the onset to a 95% block at the adductor pollicis longus muscle was reported to be 92 ± 46.9 s with 0.6 mg/kg of rocuronium and 1–3 μg/kg of fentanyl [23]. Since participants in the current study were younger and sevoflurane was used for the induction of general anesthesia, we assumed that 2 min would be enough for full relaxation of the muscles.

The current study has some unique features. First, 35% of participants had a difficult airway (CL grade III), and in two cases, intubation with direct laryngoscopy failed. This high ratio of difficult airways may contribute to the large difference (more than 10 s) of intubation time between the two groups. Most studies use “difficult intubation” which is defined as “more than two failed intubation attempts” to define a difficult airway. When “two failed intubation attempts” is a cut-off, the reported rate of difficult intubation is 5.8% in neonates and infants, 8.8% in the pediatric intensive care unit, and 14% in the neonatal intensive care unit [24,25,26]. If this criterion is applied to DL group of the current study, the rate of difficult intubation is 15.8%. In addition, the current study used CL grade to define difficult airways, and it was found that 31.5% of participants in DL had a CL grade of III. This ratio is slightly higher than the reported data, but we believe that it represents the status of the hospital where the study was conducted. It is well known that for high-quality care in the neonatal and pediatric intensive care units, the severity of the admitted neonates and infants is not light. As a result, the possibility of difficult airways is inevitably high. Second, only one experienced anesthesiologist performed all intubations. The anesthesiologist had experience using the UEscope and a direct laryngoscope more than 100 times in neonates and infants, which could greatly reduce interpersonal bias. Since the first-attempt success rate in the DL was 52.6%, a proficiency issue could be raised. However, the first-attempt success rate can vary greatly depending on the condition of the patient. In one study on pediatric difficult airway, the first-attempt success rate with direct laryngoscopy was as low as 3% [4]. Although 31.5% of participants in DL had a CL grade of III, difficult intubation (more than two failed intubation attempts) only occurred in 15.8%. This may be proof that the anesthesiologist has successfully performed tracheal intubation in difficult airway situations. Third, a picture of the glottis was taken before and after conducting the BURP maneuver. Another pediatric anesthesiologist determined the POGO score and CL grade based on the pictures, and two anesthesiologists discussed and confirmed the final result, which greatly increased the accuracy of glottis view evaluation.

## 5. Conclusions

Our findings show that the UEscope may be a better choice for tracheal intubation than conventional direct laryngoscope in neonates and infants. This finding could serve as a basis for larger, well-designed RCTs in neonates and infants with a difficult airway.

## Figures and Tables

**Figure 1 children-09-01161-f001:**
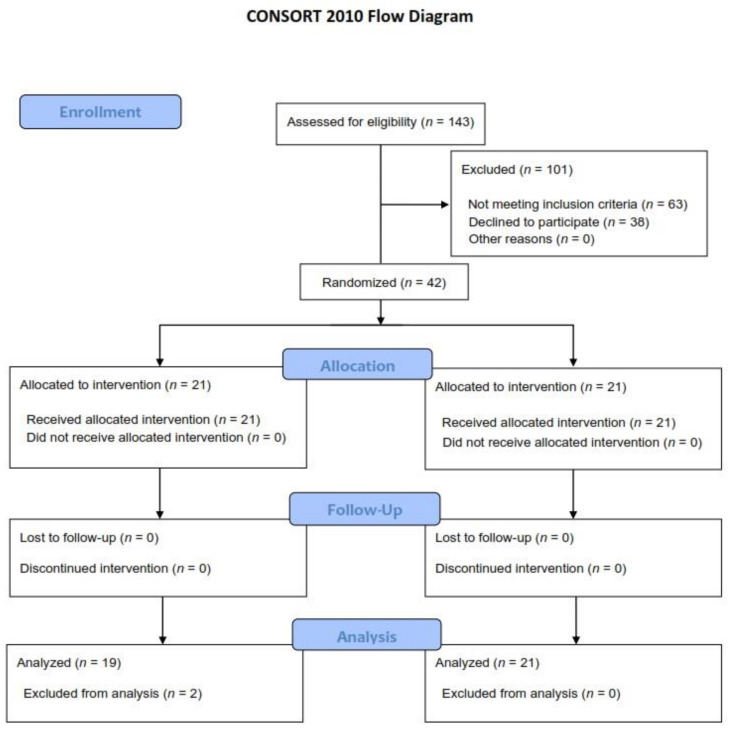
Consolidated Standards of Reporting Trials (CONSORT) diagram.

**Figure 2 children-09-01161-f002:**
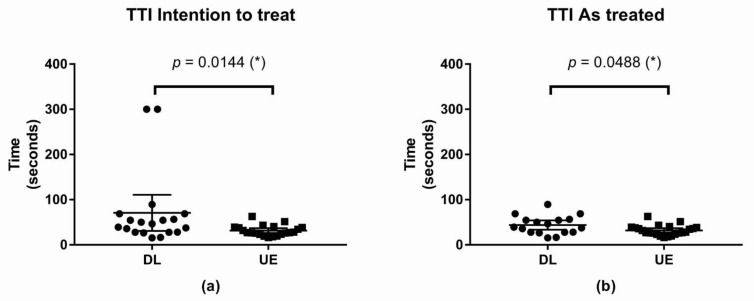
Scatter plot of Time to intubation (**a**) TTI, intention to treat (**b**) TTI, as treated; (*): *p*-value < 0.05. Abbreviations: TTI, time to intubation; DL, direct laryngoscope group; UE, UEscope group.

**Figure 3 children-09-01161-f003:**
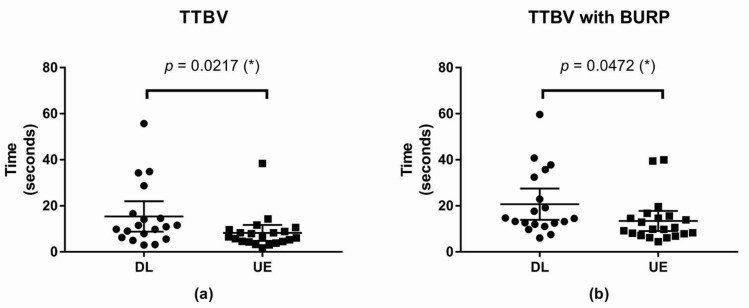
Scatter plot of Time to best view (**a**) TTBV (**b**) TTBV with the BURP maneuver; (*): *p*-value < 0.05. Abbreviations: DL, direct laryngoscope group; UE, UEscope group; TTBV, time to best view; BURP, backward, upward, rightward, and posterior pressure.

**Figure 4 children-09-01161-f004:**
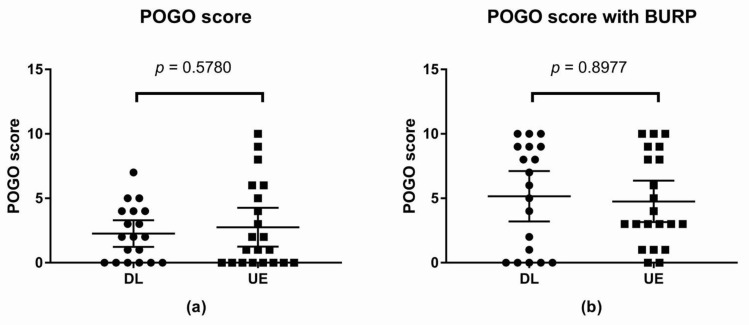
Scatter plot of the best POGO score (**a**) POGO score; (**b**) Best POGO score with the BURP maneuver. Abbreviations: DL, direct laryngoscope group; UE, UEscope group; POGO, percentage of glottis opening; BURP, backward, upward, rightward, and posterior pressure.

**Figure 5 children-09-01161-f005:**
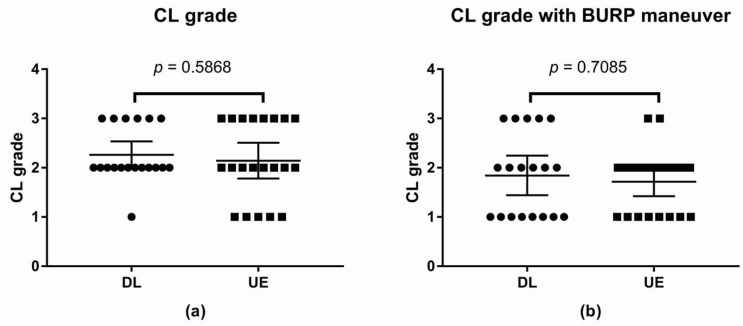
Scatter plot of Cormack and Lehane Laryngeal View grade (**a**) CL grade (**b**) CL grade with the BURP maneuver. Abbreviations: DL, direct laryngoscope group; UE, UEscope group; CL, Cormack and Lehane Laryngeal View; BURP, backward, upward, rightward, and posterior pressure.

**Figure 6 children-09-01161-f006:**
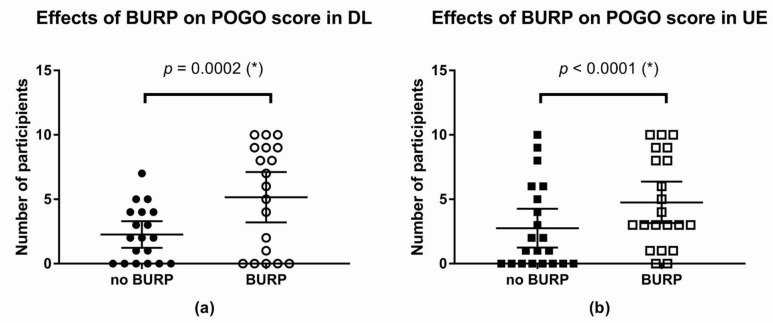
Changes in the best POGO score after the BURP maneuver in each group: (**a**) Changes in the POGO score after the BURP maneuver in DL; (**b**) changes in the POGO score after the BURP maneuver in UE. (*): *p*-value < 0.05. Abbreviations: DL, direct laryngoscope group; UE, UEscope group; POGO, percentage of glottis opening; BURP, backward, upward, rightward, and posterior pressure.

**Figure 7 children-09-01161-f007:**
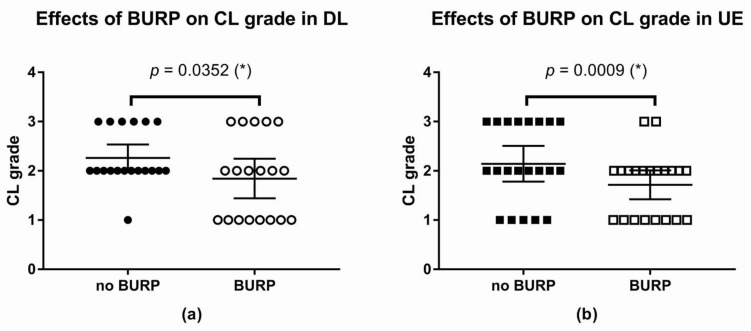
Effect of the BURP maneuver on the CL grade: (**a**) Changes in the CL grade after the BURP maneuver in the DL; (**b**) changes in the CL grade after BURP maneuver in the UE; (*): *p*-value < 0.05; Abbreviations: DL, direct laryngoscope group; UE, UEscope group; CL, Cormack and Lehane Laryngeal View; BURP, backward, upward, rightward, and posterior pressure.

**Figure 8 children-09-01161-f008:**
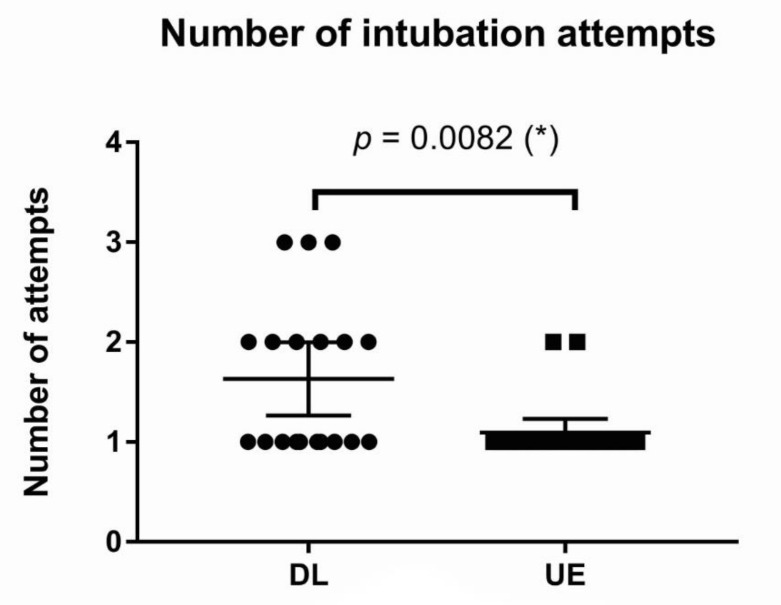
Number of intubation attempts in the DL and UE. (*): *p*-value < 0.05. Abbreviations: DL, direct laryngoscope group; UE, UEscope group.

**Table 1 children-09-01161-t001:** Demographic data ^1^.

	DL	UE
Age (days)	84 (56–113)	72 (13–110)
Neonate ^2^/Infant ^3^	4/15	6/15
Sex (male/female)	15/4	12/9
Height (cm)	58.0 (54.0–61.0)	53.8 (49.3–60.1)
Weight (kg)	5.5 (4.6–5.9)	4.3 (3.9–5.9)

^1^ Data are presented as the median (interquartile range) or number/number; ^2^ Neonate, 1–30 days old; ^3^ Infant, 31–364 days old; Abbreviations: DL, direct laryngoscope group; UE, UEscope group.

## Data Availability

The datasets generated and/or analyzed during the current study are available in the Mendeley Data repository, [https://doi.org/10.17632/wpphwkhxvj.1 accessed on 2 August 2022].

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
