# Peer review of "Comparison of the UEscope Video Laryngoscope with the Traditional Direct Laryngoscope in Neonates and Infants: A Randomized Clinical Trial"

_children, 2022, doi:10.3390/children9081161_

Round 1
Reviewer 1 Report
1) In Introduction section give a little more information about current airway management techniques in children and adults in context of your research, for exaple using this publication, please:
Matek, J.; Kolek, F.; Klementova, O.; Michalek, P.; Vymazal, T. Optical Devices in Tracheal Intubation—State of the Art in 2020. Diagnostics 2021, 11, 575. https://doi.org/10.3390/ diagnostics11030575
2) in Discussion section explain, why you have so high rate of difficult arways, please
3) in Discussion section, to do the article more comprehensive, compare more ways and devices to the pediatric airways, please
4) in Discission section give more information about superiority/inferiority in different airway technigues in the context of the EUscope, please
5) I do agree with the unique features of the study and I appreciate them, but also agree with the limitations. In my view mainly the weak cohort.
Reviewer 2 Report
In this randomized clinical trial including 40 neonates and infants, the authors compared performance of the UEscope video laryngoscope with the traditional direct laryngoscope. They showed that the time to intubation (TTI) was significantly shorter with the Uescope. This study is well performed and has the potential implications, but I have several concerns on the methodology and results.
1. This randomized clinical trial only included 40 neonates and infants, it was unclear why such a small-size was completed in tertiary hospital for 1.5 years from March 2020 to September 2021.
2. It was unclear how long time was allowed to start endotracheal intubation after application 0.6 mg/kg of rocuronium how the authors assessed whether rocuronium had achieved maximal effect before initiation of intubation.
3. In method, the authors did not specify details of preoxygenation before intubation, such as fresh gas flow, ventilation modality and preoxygenation endpoint. This would have confused the results of secondary endpoints including the incidence of desaturation and lowest SpO2 during intubation.
4. Sample size calculation was based on prior studies, but the authors did not provide the reference, and the main results of prior studies including mean and SD of TTI. Thus, reporting of sample size calculation was inadequate.
5. A single pediatric attending anesthesiologist with more than 10 years of experience in pediatric anesthesia conducted all intubations. Strangely, however, the first-attempt success rate with direct laryngoscope only was 52.6%. I am concerned that the operator was not proficient in clinical application of direct laryngoscope and use of such a low-efficient control would have biased the primary and secondary outcome in favour of the UEscope.
6. The glottic view was recorded using the built-in camera of the UEscope or bronchoscope. Is the glottic view obtained by direct laryngoscope assessed by bronchoscope? If yes, the authors should provide the details of bronchoscopic assessment the glottic view. Furthermore, I am concerned that the glottic views by bronchoscope cannot represent those obtained by direct laryngoscopy, which needs the alignment of three upper airway axes. Most important, the POGO score and CL grade were designed to assess the performance of direct laryngoscopy. There is dispution as to whether they are suitable for performance assessment of videolaryngoscopy, which does not need the alignment of three upper airway axes to provide good glottic exposure.
7. The discussion was too long, but the key points were not highlighted and the main findings were not clearly explained.
8. The contents in lines 302-312 are same as those in lines 313-327.
Round 2
Reviewer 2 Report
The authors have well addressed the questions suggested by me and I have no further question. However, we think that the authors should ask a native English speaker to improve their paper to publication standards.